# Recent Topics of Pneumococcal Vaccination: Indication of Pneumococcal Vaccine for Individuals at a Risk of Pneumococcal Disease in Adults

**DOI:** 10.3390/microorganisms9112342

**Published:** 2021-11-12

**Authors:** Nobuhiro Asai, Hiroshige Mikamo

**Affiliations:** 1Department of Clinical Infectious Diseases, Aichi Medical University Hospital, Nagakute 480-1195, Japan; nobuhiro0204@gmail.com; 2Department of Infection Control and Prevention, Aichi Medical University Hospital, Nagakute 480-1195, Japan; 3Department of Pathology, University of Michigan, Ann Arbor, MI 48109, USA

**Keywords:** pneumococcal vaccine, PCV13, PPSV23, invasive pneumococcal disease, *streptococcus pneumonia*

## Abstract

Pneumococcal disease is one of the most common and severe vaccine-preventable diseases (VPDs). Despite the advances in antimicrobial treatment, pneumococcal disease still remains a global burden and exhibits a high mortality rate among people of all ages worldwide. The immunization program of the pneumococcal conjugate vaccine (PCV) in children has decreased pneumococcal disease incidence in several countries. However, there are several problems regarding the pneumococcal vaccine, such as indications for immunocompetent persons with underlying medical conditions with a risk of pneumococcal disease, the balance of utility and cost, i.e., cost-effectiveness, vaccine coverage rate, serotype replacement, and adverse events. Especially for individuals aged 19–64 at risk of pneumococcal disease, physicians and vaccine providers should make a rational decision whether the patients should be vaccinated or not, since there is insufficient evidence supporting it. We describe this review regarding topics and problems regarding pneumococcal vaccination from the clinician’s point of view.

## 1. Introduction

Vaccination is one of the most critical public health interventions and has dramatically reduced many kinds of infectious diseases which killed or harmed infants, children, and adults. In many nations, routine immunization programs are provided for infants and children, with fewer countries providing recommendations specifically for adults. Vaccine-preventable diseases (VPDs) remain a significant health burden in infants and children as well as adults. Pneumococcal disease is one of the most common and severe VPDs. Despite the advances in antimicrobial treatments, pneumococcal disease still remains a global burden and a high mortality rate among people of all ages worldwide [1,2,3]. There are currently two different types of pneumococcal vaccines, the 13-valent pneumococcal conjugate vaccine (PCV13) and 23-valent pneumococcal polysaccharide vaccine (PPSV23), worldwide. The 7-valent pneumococcal conjugate vaccine (PCV7), which was the first vaccine approved for *S. pneumoniae*, was introduced into the routine pediatric immunization schedule in 2000 and replaced by PCV13 in 2010. In the United States of America, the incidence of PCV13 type disease has been declining to dramatically low levels among people 65 years and older through indirect effects from the pediatric PCV13 program [2]. Although PCV13 is safe and effective for preventing pneumococcal disease among children and people over 65 years, in terms of balancing the evidence and considering acceptability and feasibility, the *Advisory Committee on Immunization Practices* (ACIP) stated in June 2019 that it no longer routinely recommends PCV13 for all adults aged >65 years and instead, recommends PCV13 based on shared clinical decision-making for adults aged >65 years who do not have an immunocompromised condition [2]. Currently, the strategy of pneumococcal vaccination using PCV13 and PPSV23 differs in each country. These have made physicians and vaccine providers confused regarding vaccination. Significantly, the vaccination strategy for 19–64 years with an immunocompetent state is unclear worldwide because of the lack of evidence in this field and the absence of guidelines mentioned in detail. Nevertheless, attending doctors and vaccine providers need to make a rational decision whether the patients should have a vaccination or not. Despite this, there is insufficient evidence about the validity and efficacy of pneumococcal vaccination among immunocompetent patients with a risk of pneumococcal diseases. Additionally, the guidelines for each disease also do not mention it in detail. We describe this review from a clinician’s perspective, focusing on these topics that do not have enough evidence justifying vaccination for the patients with a risk of pneumococcal disease.

## 2. Epidemiology and Economic Burden

*Streptococcus pneumonia*, a gram-positive diplococcus (Figure 1), has more than 90 immunologically distinct serotypes, and pneumococcal disease is one of the most significant causes of mortality and morbidity worldwide [4,5].

In 2005, the World Health Organization (WHO) estimated that the disease would cause 1.6 million deaths annually, including the deaths of 0.7–1 million children aged under 5 years [6]. In advanced countries such as those in Western Europe, the USA, and Japan, *S. pneumonia* is the most common cause of community-acquired bacterial pneumonia in adults [1,2,7]. In 2007, WHO recommended that all countries include PCV in the routine infant immunization schedule. As a result, countries that introduced PCV have observed a large reduction in invasive pneumococcal disease (IPD) and pneumonia. As of March 2020, 146 countries out of 194 WHO member states have introduced PCV into their National Immunization Program either nationally or sub-nationally [7]. Despite this progress, 55% (approximately 74 million) of the global infant population not receiving PCV is problematic [8]. Noninvasive pneumococcal disease, including otitis media, sinusitis, and community-acquired pneumonia, are highly prevalent illnesses in children under 5, adults over 65, and immunocompromised patients. IPD, such as meningitis, bacteremia, and sepsis, is the most severe presentation and shows high mortality, despite the advancement of antimicrobial treatment. Septic arthritis [9], biliary tract infection [10], infective endocarditis [11], and mycotic aortic aneurysm [12] due to this pathogen have been reported, which sporadically happens. The pneumococcal vaccine is one of the most valuable methods for the prevention of pneumococcal disease.

Zhang et al. reported that by an observational study in the USA, rates of all causes of pneumonia (ACP) and IPD in patients aged 19–64 with chronic disease and immunocompromised conditions were significantly higher than healthy adults. The rates of ACP and IPD in adults of 19–64 years with chronic disease were 3.6 and 4.6 times higher in healthy adults, respectively, and 5.3 and 10.5 times higher for those with immunocompromised conditions. If the patients had >2 medical conditions, the rates of ACP and IPD were 8.1 and 10.6 times higher for those with chronic disease, and 6.3 and 13.4 times higher for those with immunocompromised conditions in comparison with healthy adults. In addition, medical costs per episode of ACP in the patients with chronic disease and immunocompromised conditions were 6534 and 9168 USD, which were more expensive than those in healthy adults of 4725 USD. Pneumococcal disease in patients aged 19–64 with chronic disease and immunocompromised conditions can lead to a substantial economic burden that should be prevented [13].

## 3. Type of Pneumococcal Vaccine

Currently, we have two different types of pneumococcal vaccines licensed (shown in Table 1), which are the PPSV23 (serotypes 1, 2, 3, 4, 5, 6B, 7F, 8, 9N, 9V, 10A, 11A, 12F, 14, 15B, 17F, 18C, 19A, 19F, 20, 22F, 23F, 33F) and PCV13 (serotypes 1, 3, 4, 5, 6A, 6B, 7F, 9V, 14, 18C, 19A, 19F, 23F). PPSV23 induces a T-cell–independent, exclusively humoral immune response resulting in pneumococcal serotype (PS)-specific antibody formation. PPSV23, the first vaccine commercialized, is poorly immunogenic in children under 2 years and neither develops immunological memory nor eliminates the carrier state [14,15,16].

Therefore, this has not been recommended for infants and children aged under 2 years. On the contrary, PCV comprises pneumococcal capsular polysaccharides covalently linked to a protein, inducing a T-cell—dependent immune response, contributing to the formation of both PS-specific antibodies and memory B cells, thus establishing immunological memory, which is considered as an important correlate of vaccine effectiveness and long-lasting protection [14,15,16]. In fact, a meta-analysis and systematic review showed that a single dose of PCV13 elicits a better immune response among adults compared with PPSV23 while having a similar safety profile to PPSV23 [17]. For these reasons, PCV13 is being recommended for infants and children aged under 2 years. As described later, the guidelines for hematopoietic cell transplantation (HCT) mention that patients who receive HCT should be vaccinated with PCV13 [18]. The first PCV commercialized was the 7-valent PCV, replaced later by the 10-valent PCV (PCV10) and PCV13, which are currently used worldwide. It has been reported that PCV13 is effective not only against IPD but also against pneumonia and affects nasopharyngeal carriage, providing herd immunity to the elderly population [14,15,16,17]. As for the immunogenicity, previous studies showed that PCV13 could induce a more excellent functional immune response than PPSV23 for the majority of serotypes covered by PCV13, suggesting that PCV13 could offer immunological advantages over PPSV23 for the prevention of vaccine-type pneumococcal infection as well as safety among people aged 65 years and older [17]. Nevertheless, considering the cost-effectiveness, the routine immunization program of PCV13 for people aged 65 and older is still pending in Japan since 2014. Although ACIP used to recommend routine immunization of PCV13 for people aged 65 years and older, it currently no longer does it [1]. European countries such as the UK, France, and Germany have the same strategy [19,20,21].

## 4. Vaccine Schedule and Interval

For patients who need both pneumococcal vaccines, ACIP recommends that PCV13 is given first, followed by PPSV23 more than 1 year after PCV13 [1]. If PCV has been given, PPSV23 should be given more than 1 year after PCV13 and more than 5 years after any PPSV23. If PPSV23 is previously given, PCV13 should be administered at least 1 year after PPSV23. If patients are in special situations such as having immunocompromised conditions, cochlear implant, or CSF leak (Table 2), they should have 1 dose PCV13 followed by 1 dose PPSV23 at least 8 weeks later, then another dose of PPSV23 at least 5 years after the previous PPSV23.

Sequential vaccination of PCV13 prior to PPSV has the potential to enhance the immune response to the polysaccharide vaccine by stimulating memory B cells and priming the immune system [14,15,16]. Jackson et al. performed a randomized clinical trial to compare the immunogenicity and safety of sequential vaccination between PCV13 prior to PPSV23 (PCV13/PPSV23), PCV13/PCV13, and PPSV23/PPSV23 group in pneumococcal vaccine naïve 60–64 years adults. They measured anti-pneumococcal opsonophagocytic activity (OPA) titers before and 1 month after each vaccination. The results showed that the OPA titers in the PCV13/PPSV23 group exhibited significantly higher levels than PPSV23/PPSV23 group, even though the OPA titers declined from 1 month to 1 year after PCV13 administration but remained higher than pre-vaccination baseline titers [16,22]. Grenberg et al. conducted a randomized, modified double-blind study in 720 pneumococcal vaccine-naïve adults 60–64 years of age. Subjects received either PCV13 at year 0 and PCV13 at year 1; PCV13 at year 0 and PPSV23 at year 1; or PPSV23 at year 0 and PCV13 at year 1. An initial PCV13 increased the anti-pneumococcal response to subsequent administration of PPSV23 for many of the serotypes in common to both vaccines. In contrast, an initial PPSV23 resulted in a diminished response to subsequent administration of PCV13 for all serotypes. Neither group exhibited severe adverse events [23]. Hammitt et al. reported that repeat re-vaccination with PPSV23 with a 2nd, 3rd, or 4th dose administered 6 or more years after the prior dose was immunogenic and generally well-tolerated in people aged 55–74 years [24].

## 5. Indications for Immunocompromised Patients

### 5.1. HIV Infected/AIDS

A recent study showed that the incidence rate of pneumococcal disease among patients with human immunodeficiency virus (HIV) infection is 190 per 100,000 patient-years of follow-up, much higher than 38 per 100,000 in the general population [25]. In the study, 224 of the 324 patients (69%) had started Combination Antiretroviral Therapy with a median CD4 T-cell count of 440 cells/mm3 at the diagnosis of pneumococcal disease. Pneumococcal disease among patients with HIV infection often requires hospitalization, and the mortality rates range up to 25% [26]. While ACIP recommends that patients with HIV infection should have pneumococcal vaccination with PCV13, followed 8 weeks later by PPSV23 [1], the clinical efficacy of this combined vaccination schedule is unclear. A systematic review and meta-analysis documented the clinical efficacy of PCV13/PPSV23 among HIV-infected patients [27]. Delaying vaccination until CD4 T-cell count above 200 cells/mm^3^ is beneficial for vaccination response and should be recommended [27,28].

### 5.2. Hematologic and Non-Hematologic Malignancy

The American Society for Blood and Marrow Transplantation/European Society for Blood Marrow Transplantation (ASBMT/EBMT) guidelines recommend patients who receive hematopoietic transplantation receive three sequential doses of PCV13 starting 3 to 6 months after hematopoietic cell transplantation (HCT). The fourth dose of PPSV-23 is recommended 8 weeks after the third dose of PCV13 [18]. In contrast, the guidelines for non-hematologic malignancy do not mention the strategy for preventing pneumococcal disease. It was previously reported that few cancer patients were vaccinated [29]. Urun et al. reported that the overall vaccination rate for cancer patients was 17% and 4.2% for influenza and pneumococcal vaccination, respectively [30]. They interviewed, in person, 296 unvaccinated patients with malignancy. The leading causes for not getting vaccinated were lack of knowledge for indication by the patients (33.5%), getting chemotherapy (22.1%), fear of side effects (12.5%), lack of efficacy (12.1%), and not being advised by the attending physicians (5.9%) [31]. Considering the vaccine’s side effects, if attending physicians have enough knowledge about vaccination, most cancer patients could have been vaccinated before starting chemotherapy. It would be necessary to educate oncologists for the broader spreading of vaccination among cancer patients. The guideline by the Infectious Disease Society of America (IDSA) and ACIP states that cancer patients should receive the sequential PCV13 and PPSV23 immunization. The timing should be at least 2 weeks prior to starting chemotherapy [1,32]. Despite this, it is common for cancer patients receiving chemotherapy to miss the occasion to be vaccinated. Some physicians might worry about the efficacy and safety of vaccination after starting chemotherapy. Choi conducted a randomized control trial to evaluate the optimal timing of administering PCV13 among gastric and colorectal cancer patients. Patients with gastric and colorectal cancer had received adjuvant chemotherapy (Fluoropyrimidine or Fluoropyrimidine plus Oxaliplatin). The overall antibody response to PCV13 was adequate in the patients, and no significant differences were found when they were vaccinated two weeks prior or on the day of chemotherapy initiation [31]. Although there is little evidence of optimal timing getting vaccination, the day of chemotherapy initiation could be acceptable for patients with non-hematological malignancy to get vaccinated.

### 5.3. Chronic Kidney Disease

Having learned the guidelines regarding kidney disease in the US, Japan recommends that chronic kidney disease (CKD) patients should be vaccinated [1,32]. Several studies showed the immunogenicity and safety of PCV7, or PPSV23, among children and adults with CKD [33,34]. As for patients with hemodialysis, Pladys conducted a retrospective cohort in France and reported that sequential vaccination of PCV13 and PPSV23, or only PCV13 but not with PPSV23, is correlated with a decrease of one year-mortality in patients who start HD [35]. They insisted that the end-stage renal disease (ESRD) patients who start HD should be vaccinated with PV13 followed by PPSV23. Further, Mitra reported that PCV13 could induce antibody response to vaccine serotype in patients with ESRD and on dialysis at 2 months post-vaccination. However, the antibody concentrations at 12 months post-vaccination decreased, and further analysis was not possible because of the too-small sample size of the study [36].

### 5.4. Autoimmune Inflammatory Diseases

It is well-known that patients with autoimmune inflammatory diseases (AIID) have a risk of pneumococcal disease [37]. The European League against Rheumatism (EULAR) recommendations agree with ACIP guidelines that routine vaccination with PCV13 and PPSV23 for adults aged >19 years with AIID [38]. However, the pneumococcal vaccine coverage rates among AIID patients remain low, showing a percentage of 6 to 62% [39,40]. The main reason for not being vaccinated among the patients is a fear of adverse events and physicians’ concern that treatment with a biological agent can be associated with an impaired immune response to vaccines [41]. Richi et al. assessed the efficacy on the immune response after pneumococcal vaccination among AIID patients who received a biological agent by using functional antibodies by an opsonophagocytosis killing assay (OPKA) [42]. They showed that AIID patients treated with biological agents had a functional antibody phagocytic response after pneumococcal vaccination using PCV13 and/or PPSV23.

## 6. Recommendation for Immunocompetent Persons with Underlying Medical Conditions

### 6.1. Heart Disease

Since chronic heart disease is a risk factor for pneumococcal disease, the guideline recommends that patients with heart disease should be vaccinated with a pneumococcal vaccine [43]. A systematic review suggests that pneumococcal vaccination is associated with decreased risk of all-cause mortality in patients at very high cardiovascular risk [44]. Additionally, the potential benefit of pneumococcal vaccination in patients at very high risk of cardiovascular disease is based on the eviction of the exposure to *Streptococcus pneumoniae* antigens that are thought to be immunologically similar to oxidized LDL cholesterol which may increase atherosclerotic plaques, as was demonstrated in animal studies [45,46]. In addition, reductions in pneumococcal-associated disease incidence also prevent potentially pernicious systemic effects such as sympathetic activation, tissue hypoxemia, the release of pro-inflammatory cytokines with endothelial dysfunction, and hypercoagulability [47]. Takotsubo cardiomyopathy, a condition triggered by a significant stressor, such as hospitalization for pneumonia, may also explain some of the increased cardiovascular death associated with pneumococcal disease [48]. Recently, the mechanism that *S. pneumoniae* directly inhibits cardiac contractility was reported, which was that the pneumococcus is capable of translocating into the heart and forms discrete, non-purulent cardiac microlesions that may directly disrupt contractility.

### 6.2. Chronic Obstructive Pulmonary Disease (COPD)

Most supportive evidence is collected for COPD to justify the vaccination among patients with chronic respiratory disease. COPD is a risk factor for severe pneumonia and IPD [49]. PCV13 effectively prevented vaccine-type pneumococcal, bacteremic, and nonbacteremic community-acquired pneumonia and vaccine-type invasive pneumococcal disease but not in preventing community-acquired pneumonia from any cause [50]. A meta-analysis documented that PPSV23 effectively reduced the incidence of pneumonia and acute exacerbation among COPD patients [50]. The guidelines for COPD concluded that injectable polyvalent pneumococcal vaccine provides significant protection against community-acquired pneumonia, although no evidence indicates that vaccination reduced the risk of pneumococcal pneumonia. It is inconclusive whether all severe COPD patients should have a pneumococcal vaccine. PPSV23 has been shown to reduce the incidence of community-acquired pneumonia in those under 65 years of age with COPD with FEV1.0 <40% [51]. This suggests that a severe stage of COPD (GOLD 3–4) should be vaccinated, especially when the patients are <65 years. Combined vaccination of influenza vaccine and PPSV23 is effective in the prevention of acute exacerbation of COPD. However, the additive effect of PPSV23 and influenza vaccine was found during the first year, not during the second year after vaccination. More analysis would be needed to conclude the additive efficacy of the combined vaccination [52]. Pneumococcal vaccination could be effective in the prevention of post-influenza infection [53]. To prevent post-influenza infection, combined vaccination of pneumococcal vaccine and influenza vaccine could be an optimal preventive method.

### 6.3. Asthma

“Adults with asthma are greater risk of pneumococcal infection, with an estimated population attributable risk between 12% and 17% for adults with asthma” [54]. In addition, inhaled corticosteroid (ICS) use has been associated with an increased risk of pneumonia in asthmatics patients [55], even though ICS is an optimal asthma treatment. Thus, ACIP recommends that asthma patients should have a pneumococcal vaccination. On the contrary, Global Initiative for Asthma (GINA) documented insufficient evidence to recommend that the patients have the vaccine, although people with asthma, particularly children and elderly, are at risk of IPD [56,57,58]. Additionally, no one knows what severity of asthma patients should have the vaccine. Asthma patients are expected to show a lower antibody response after vaccination with PPSV23 [59]. Some documented that asthma, a Th-2 predominant immune environment, suppresses the humoral immune response to pneumococci in humans and mice [60]. Moreover, it has been reported that children with eczema have inadequate pneumococcal antibody response due to Th-2 predominant immune environment [61]. We need to reconsider the current vaccination strategy for asthmatics patients by asthma status.

### 6.4. Interstitial Lung Disease

While interstitial lung disease (ILD) is one of the most severe respiratory diseases, ATS/ERS/JRS/ALAT guideline or ACIP guideline never mentions whether a pneumococcal vaccine is indicated for the patients with ILD [1,62]. There is the possibility that pneumococcal vaccination is not effective among patients with ILD. It is because patients with ILD are commonly using corticosteroids and/or immunosuppressive agents. Akamatsu et al. conducted a study that evaluating the efficacy of PPSV 23 among patients with ILD. They reported that anti-pneumococcal antibody levels following vaccination were similar to the control group (who were not being treated with immunosuppressive therapy). They concluded that PPSV23 is effective in the patients, even though they are undergoing systemic steroid and/or immunosuppressive therapy [63]. In addition, Kuronuma et al. documented that both PCV13 and PPSV23 induced anti-pneumococcal antibodies in patients with ILD receiving systemic steroid and/or immunosuppressive therapy [64]. Generally, ILD is more severe and tends to be worse than asthma, and some ILD patients are complicated when they also have COPD.

### 6.5. Vaccination Coverage Rate among Patients with Any Risk of Pneumococcal Disease

The vaccination coverage rate (VCR) is quite different in each country. In Japan, although a routine immunization program of PPSV23 for adults 65 years or older started in October 2014, the VCR of the pneumococcal vaccine among them is 33% [65]. We have no data regarding the VCR of the pneumococcal vaccine among patients 19–64 years who are in immunocompetent states, such as liver cirrhosis, asthma, ILD, or heart disease. The VCR of the pneumococcal vaccine among patients 18–64 at any risk of pneumococcal disease in the UK is 32% [66]. In the USA, it was reported that the VCR of the pneumococcal vaccine among adults aged 19–64 years with risk was 24.5% in 2017, while that among those aged ≥65 years was 69.0% in 2017 [67]. In this report by Mayo Clinic, a vaccine was given by a specialist in the clinic more frequently than in hospitals. For the pneumococcal vaccine to spread more widely, we need to analyze why its VCR does not increase and develop a strategy.

### 6.6. Serotype Replacement

The introduction of PCVs among children in many countries had decreased the incidence of IPD, pneumococcal pneumonia, otitis media, and meningitis, while the serotype replacement of pneumococcal strain has been seen in several areas and countries [68,69,70]. The increased nonvaccine serotypes are different in each country; 6C, 15A/B/C, 23A, and 35B in the United States [71,72], 15A and 23B in Norway [73] and Germany [74], and 12F, 15A, 24F, and 35B in France [75]. These are difficult to explain by serotype replacement.

A 15-valent pneumococcal conjugate vaccine (PCV15), which includes all components of PCV13 plus polysaccharide conjugates of two additional serotypes (22F and 33F), has the potential to address critical medical and public health needs by providing broader coverage for leading serotypes associated with pneumococcal disease worldwide. The two PCV15 unique serotypes contribute to causing IPD in children and adults following widespread use of PCV13 in children in many countries, likely due to their invasiveness capacity [76,77,78,79]. By 2013 in the US, residual IPD caused by serotype 22F among children <5 years and adults ≥18 years were 11% and 13%, respectively, while serotype 33F caused 10% and 5% of residual IPD cases in children <5 years and adults ≥18 years, respectively [79]. Some trials had already shown the safety, tolerability, and immunogenicity for the serotypes in 18–45 [80] and >50 aged adults [81]. A 20-valent pneumococcal conjugate vaccine (PCV20), which is anticipated to provide more robust protection than PPSV23 for the serotypes in common with PCV20, includes all components of PCV13 plus polysaccharide conjugates of seven additional serotypes (8, 10A, 11A, 12F, 15B, 22F, and 33F) [82]. Six of the seven PCV20 unique serotypes (8, 10A, 12F, 15B, 22F, and 33F) have emerged as prominent nonvaccine serotypes responsible for IPD worldwide [79]. In addition, PCV20 unique serotypes are also responsible for a substantial proportion of IPD-related deaths. Oligbu et al. reported that PCV20 unique serotypes (8, 11A, 15B, 15C, 22F, and 33F) were responsible for 73% to 92% of IPD-related deaths in children after PCV13 introduction in England and Wales [83]. The phase 2 trial evaluated the safety, tolerability, and immunogenicity of PCV20 among adults aged 60–64 without any prior pneumococcal vaccination had been performed. It was reported that PCV 20 was well-tolerable in adults aged 60–64 with a safety profile [82]. A phase 1 trial of PCV20 in healthy Japanese adults of 18–49 aged showed that PCV20 was tolerable with a safety profile and enough immunogenicity for the seven additional serotypes [84]. Both PCV15 and PCV20 could expand the protection of pneumococcal disease. However, it is unclear whether these PCVs will solve the problem of serotype replacement, even if they are promising. A new type of vaccine, which is serotype-independent, should be needed. Protein antigens that are conserved across pneumococcal serotypes may offer an alternative to serotype-restricted vaccination strategies [85]. Currently, some trials regarding protein-based vaccines such as pneumolysin and pneumococcal histidine triad protein D for the protection of pneumococcal disease are ongoing [86].

### 6.7. Adverse Events by Vaccination

Every vaccination could cause adverse events; however, it is not a reason that people hesitate to be vaccinated, except for an anaphylaxis reaction. The most common adverse event of PCV13 is pain in 44.5%, followed by erythema in 27.6%, and swelling in 21.0%. For all, 55.6% of the patient getting a shot has some adverse events, which are focal and not severe [87]. Currently, immunization programs using mRNA COVID-19 vaccines have started in the USA. People get vaccinated with mRNA SARS-CoV-2 vaccine. A total of 0.2% of adverse events has been reported. Anaphylaxis reaction happens in only 21/1,000,000 injections (0.000021%). Of those, 17 of 21 had a medical history of drug allergy or anaphylaxis reaction [88]. This result reproduced the previous report [89]. The probability of a severe adverse event associated with vaccination is 0.0007–0.001% [90]. This probability is similar to the odds that a flash of lightning would hit people (0.0001%), or a meteorite could hit and kill people (0.0006%) [91,92]. It would be strange to hesitate to get a vaccine for such a low probability of adverse events. Clinicians should be aware of this fact and provide patients requiring a vaccine with correct information.

## 7. Discussion

Pneumococcal disease is causing some kinds of infections such as pneumonia, meningitis, bacteremia, infective endocarditis, etc. [9,10,11,12,93]. Pneumococcal disease is one of the most common VPD, causing a life-threatening in people of all ages. In fact, after the introduction of PCV7 or 13, rates of IPD due to these serotypes are associated with a significant decline in several areas and countries [2,94]. In addition, some documented by a systematic review and a meta-analysis that the efficacy and safety of both PCV13 and PPSV23 are effective for preventing pneumococcal disease compared to no vaccination and showing a safe profile in children and older people [95]. Despite these, one of the reasons for not prevailing with the pneumococcal vaccines is that it is costly by itself. ACIP used to recommend a sequential vaccination PCV13 after PPSV23; however, it no longer recommends it because the immunization program in all aged people is not cost-effective [2]. Neither does Japan. The sequential vaccination is cost-effective for preventing pneumococcal disease in 65 years aged people due to the lack of evidence to support it. We evaluate the cost-effectiveness of the pneumococcal vaccine by QALY [96] and ICER [97]. The QALY and ICER are essential concepts in cost-effectiveness analysis. QALY is used to estimate the quality of life (QOL) and life years and can be used to compare the efficacies of cancer and cardiovascular treatments. ICER is defined as the difference in cost between treatments divided by the differences in their effects, with a smaller ICER indicating better cost-effectiveness [98]. We are concerned whether the pneumococcal immunization program is introduced based on the cost-effectiveness studies evaluated by QALY and ICER. There are several studies published for the cost-effectiveness of pneumococcal vaccine by QALY and ICER in different countries [21,99,100]. However, the results differ in each area and country. The cost-effectiveness conclusions regarding the pneumococcal vaccine are variable due to a different vaccine cost and a different vaccine strategy in each country. No one knows how much ICER or QALY is appropriate to support a routine immunization program. In addition, starting the routine immunization program of PCV13 among children in several countries, serotype replacement has been confirmed as describes above. The cost-effectiveness and efficacy might be changing year by year, and the cost-effectiveness data should be analyzed, and the vaccine strategy should be reconsidered in the near future.

Surprisingly, the VCR of the pneumococcal vaccine is much lower than we expected [36]. Unfortunately, there are no data regarding VCR among patients aged 19–64 years, so we could not find why the pneumococcal vaccine has not prevailed widely. We need a further study to analyze this in the population. The potential solutions to improve the low VCR in the population include not only an education to the patients about the importance of vaccination but also reminders for health care professionals, as well as follow-up protocols with patients to make sure they have completed the vaccines.

We believe that sequential vaccination and PCV13 and PPSV23 would be the best vaccination strategy for preventing pneumococcal disease among adults aged 19–64 with a risk of pneumococcal disease, even though we have cost-effective issues regarding vaccination. Current all pneumococcal vaccines are serotype-dependent. Thus, concerns regarding serotype replacement would remain, even if both results of trials in PCV15 and PCV20 are promising. Effective serotype-independent vaccines such as protein-based vaccines should be necessary.

## 8. Conclusions

Pneumococcal vaccines, PCV13 and PPSV23, effectively prevent pneumococcal disease among adults aged 19 and older. Sequential vaccination of PCV13 and PPSV23 might be a preventive option among patients with middle to severe stage of immunocompetent disease, even if there is insufficient evidence supporting it. More study analyzing post-vaccination prognosis is needed. Although pneumococcal disease is one of the most common VPDs, showing a high mortality rate, the low VCR is another concern to prevent pneumococcal disease. We should make a follow-up protocol for the patients and vaccine providers to ensure whether vaccines are provided appropriately.

## Figures and Tables

**Figure 1 microorganisms-09-02342-f001:**
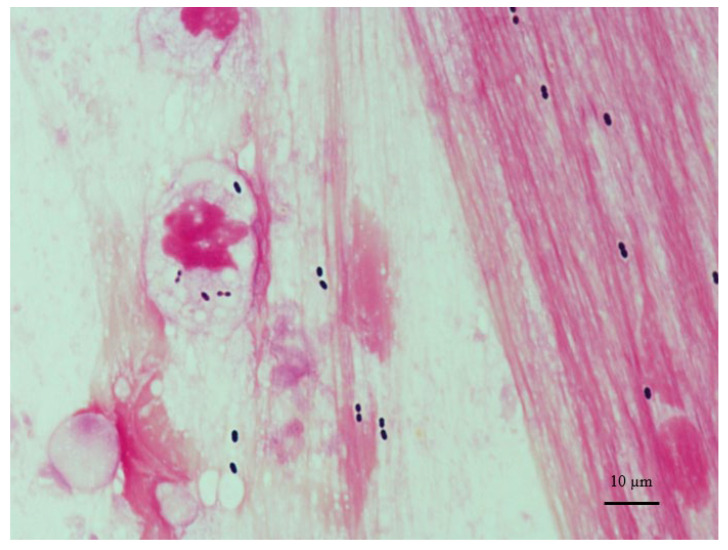
Shows gram-positive cocci on gram stain by sputum culture, detecting *Streptococcus pneumonia.* The picture was provided by Tomoko Ohno and Hiroyuki Suematsu, who are microbiologists at Aichi Medical University hospital (Original magnification ×1000).

**Table 1 microorganisms-09-02342-t001:** Comparison of serotypes covered in both vaccines.

Type of Pneumococcal Vaccine	PCV13	PPSV23
Serotypes included	1, 3, 4, 5, 6A, 6B, 7F, 9V, 14, 18C, 19A, 19F, 23F	1, 2, 3, 4, 5, 6B, 7F, 8, 9N, 9V, 10A, 11A, 12F, 14, 15B, 17F, 18C, 19A, 19F, 20, 22F, 23F, 33F

Serotypes with underline drawing are covered in both vaccines.

**Table 2 microorganisms-09-02342-t002:** Medical indication group.

**Immunocompetent Persons**	
Alcoholism Chronic liver disease Chronic lung disease Chronic heart disease	Cigarette smoking Diabetes mellitus Cochlear implant CSF leak
**Immunocompromised Persons**	
Congenial or acquired asplenia Sickle cell disease/other hemoglobinopathies Chronic renal failure Congenial or acquired immunodeficiencies Generalized malignancy HIV infection Hodgkin disease	Iatrogenic immunosuppression Leukemia Lymphoma Multiple myeloma Nephrotic syndrome Solid organ transplant

## Data Availability

This report did not have any data.

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
