# Peer review of "Recent Topics of Pneumococcal Vaccination: Indication of Pneumococcal Vaccine for Individuals at a Risk of Pneumococcal Disease in Adults"

_microorganisms, 2021, doi:10.3390/microorganisms9112342_

Round 1
Reviewer 1 Report
This review article is well written and provides significant topics of pneumococcal infection and vaccination. I only ask the authors for brief additions and minor corrections as listed below.
- In 1. Introduction section (page 2 of 13), the following two sentences mean same, therefore, please modified these sentences.
Pneumococcal vaccination strategy differs in each country.
Currently, the strategy of pneumococcal vaccination using PCV13 and PPSV23 differs in each country.
- The current pneumococcal vaccines such as PCV13 and PPSV23 compose of capsular polysaccharides as described in 3. Type of pneumococcal vaccine (page 3 of 13). If possible, please describe about the immunological characteristics of pneumococcal capsular polysaccharides (e.g., T cell independent antigen), the immune responses (e.g., T cell response, IgG subtypes, memory response), and the mechanism by which these antigens prevent pneumococcal infection/diseases. These information will help readers better understand this review.
- If possible, I think this article would be better to have information and/or discussion on future prospect for pneumococcal vaccination. For examples, development of new vaccines such as pneumococcal surface antigen is progressing, will it be a breakthrough in solving the problems of current vaccines?
Author Response
Manuscript ID: microorganism- 1315860      Date: Aug 10, 2021
Title: Recent topics of pneumococcal vaccination: Indication of pneumococcal vaccine for individuals at a risk of pneumococcal disease in adults
Thank you very much for your great reviewing. We totally agree with your opinions and made appropriate revisions according to your suggestions. The revised portion was red in the text. We strongly believe that this revision made the text better than before. Please, consider my article for publication in your journal.
Sincerely,
Nobuhiro Asai and Hiroshige Mikamo,
Reviewer 1
Comments and Suggestions for Authors
This review article is well written and provides significant topics of pneumococcal infection and vaccination. I only ask the authors for brief additions and minor corrections as listed below.
- In 1. Introduction section (page 2 of 13), the following two sentences mean same, therefore, please modified these sentences.
Pneumococcal vaccination strategy differs in each country.
Currently, the strategy of pneumococcal vaccination using PCV13 and PPSV23 differs in each country.
→I am so sorry that these sentences were miswritten. I re-wrote it down as follows. Thank you so much for noticing it.
In page 4
recommends PCV13 based on shared clinical decision-making for adults aged>65years who do not have an immunocompromised condition [2]. Currently, the strategy of pneumococcal vaccination using PCV13 and PPSV23 differs in each country. These have made physicians and vaccine providers confused regarding vaccination. Significantly, the vaccination strategy for 19-64 years with an immunocompetent state is unclear worldwide because of the lack of evidence in this field or the absence of guidelines mentioned in detail. Nevertheless, attending doctors and vaccine providers need to make a rational decision whether the patients should have a vaccination or not. Despite this, there is insufficient evidence about the validity and efficacy of pneumococcal vaccination among immunocompetent patients with a risk of pneumococcal diseases. Besides, the guidelines for each disease also DO NOT mention it in detail. We describe this review from a clinician’s perspective, focusing on these topics that do not have enough evidence justifying vaccination for the patients with a risk of pneumococcal disease.
- The current pneumococcal vaccines such as PCV13 and PPSV23 compose of capsular polysaccharides as described in 3. Type of pneumococcal vaccine (page 3 of 13). If possible, please describe about the immunological characteristics of pneumococcal capsular polysaccharides (e.g., T cell independent antigen), the immune responses (e.g., T cell response, IgG subtypes, memory response), and the mechanism by which these antigens prevent pneumococcal infection/diseases. These information will help readers better understand this review.
→I totally agree with your opinion. I added the sentences as follows.
In page 5
Type of pneumococcal vaccine
Currently, we have two different types of pneumococcal vaccines licensed (shown in Table 1), which are PPSV23 (serotypes 1, 2, 3, 4, 5, 6B, 7F, 8, 9N, 9V, 10A, 11A, 12F, 14, 15B, 17F, 18C, 19A, 19F, 20, 22F, 23F, 33F) and PCV13 (serotypes 1, 3, 4, 5, 6A, 6B, 7F, 9V, 14, 18C, 19A, 19F, 23F). PPSV23 induces a T-cell–independent, exclusively humoral immune response resulting in pneumococcal serotype (PS)-specific antibody formation. PPSV23, the first vaccine commercialized, is poorly immunogenic in children under 2 years and neither develops immunological memory nor eliminates the carrier state [14-16].
Table 1. Comparison of serotypes covered in both vaccines
|
Type of pneumococcal vaccine |
PCV13 |
PPSV23 |
|
Serotypes included |
1, 3, 4, 5, 6A, 6B, 7F, 9V, 14, 18C, 19A, 19F, 23F |
1, 2, 3, 4, 5, 6B, 7F, 8, 9N, 9V, 10A, 11A, 12F, 14, 15B, 17F, 18C, 19A, 19F, 20, 22F, 23F, 33F |
Serotypes with underline drawing are covered in both vaccines.
Therefore, this has not been recommended for infants and children aged under 2 years. On the contrary, PCV comprises pneumococcal capsular polysaccharides covalently linked to a protein, inducing a T-cell–dependent immune response, contributing to the formation of both PS-specific antibodies and memory B cells, thus establishing immunological memory, which is considered as an important correlate of vaccine effectiveness and long-lasting protection [14-16].
- If possible, I think this article would be better to have information and/or discussion on future prospect for pneumococcal vaccination. For examples, development of new vaccines such as pneumococcal surface antigen is progressing, will it be a breakthrough in solving the problems of current vaccines?
→Since I agree with your suggestion, the following sentences were added. I appreciate your great reviewing and feedback.
In page 23
Surprisingly, the VCR of the pneumococcal vaccine is much lower than we expect [36]. Unfortunately, there are no data regarding VCR among patients aged 19-64 years, so we could not find why the pneumococcal vaccine has not prevailed widely. We need a further study to analyze this in the population. The potential solutions to improve the low VCR in the population include not only an education to the patients about the importance of vaccination but also reminders for health care professionals, as well as follow-up protocols with patients to make sure they have completed the vaccines, and serologic tests are routinely checked and repeated when appropriate.
We believe that sequential vaccination and PCV13 and PPSV23 would be the best vaccination strategy for preventing pneumococcal disease among adults aged 19-64 with a risk of pneumococcal disease, even though we have cost-effective issues regarding vaccination. Current all pneumococcal vaccines are serotype-dependent. Thus, concerns regarding serotype replacement would remain, even if both results of trials in PCV15 and PCV20 are promising. Effective serotype-independent vaccines such as protein-based vaccines should be necessary.

Reviewer 2 Report
The current article by Nobuhiro and Hiroshige, reviews the current literature on the recent topics of pneumococcal vaccination: indication of pneumococcal vaccine for individuals at a risk of pneumococcal disease in adults.The title of the paper is in line with the body of the manuscript.The topic is current and timely and very important for the world's scientific community in the current period of the global COVID-19 pandemic. The authors have written a clear and detailed review and the material is well presented, although the authors do not express particular personal ideas on the future prospects for investigation. The references used are suitable and it is new and updated material, I believe that the article can be accept pending other revision, even if I have below some observations that I think they can make the paper more complete:
-
The authors should include in their discussion and their conclusions greater personal opinion
-
In the conclusions the authors should include suggestions on how to set up new immunization programs
Author Response
Manuscript ID: microorganism- 1315860      Date: Aug 10, 2021
Title: Recent topics of pneumococcal vaccination: Indication of pneumococcal vaccine for individuals at a risk of pneumococcal disease in adults
Thank you very much for your great reviewing. We totally agree with your opinions and made appropriate revisions according to your suggestions. The revised portion was red in the text. We strongly believe that this revision made the text better than before. Please, consider my article for publication in your journal.
Sincerely,
Nobuhiro Asai and Hiroshige Mikamo,
Reviewer 2
- The authors should include in their discussion and their conclusions greater personal opinion
- In the conclusions the authors should include suggestions on how to set up new immunization programs
→I agree with your opinion. Thus, I added the following sentences. I really apricate your great reviewing and feedback for us.
In page 23
Surprisingly, the VCR of the pneumococcal vaccine is much lower than we expect [36]. Unfortunately, there are no data regarding VCR among patients aged 19-64 years, so we could not find why the pneumococcal vaccine has not prevailed widely. We need a further study to analyze this in the population. The potential solutions to improve the low VCR in the population include not only an education to the patients about the importance of vaccination but also reminders for health care professionals, as well as follow-up protocols with patients to make sure they have completed the vaccines, and serologic tests are routinely checked and repeated when appropriate.
We believe that sequential vaccination and PCV13 and PPSV23 would be the best vaccination strategy for preventing pneumococcal disease among adults aged 19-64 with a risk of pneumococcal disease, even though we have cost-effective issues regarding vaccination. Current all pneumococcal vaccines are serotype-dependent. Thus, concerns regarding serotype replacement would remain, even if both results of trials in PCV15 and PCV20 are promising. Effective serotype-independent vaccines such as protein-based vaccines should be necessary.

Reviewer 3 Report
This in an interesting article by Asai and Mikamo that review the indications of pneumococcal vaccination for individuals at risk of pneumococcal infection. However, it needs some changes and explanations to improve its quality. For this reviewer the main lack is that when authors review the indications in immunosuppressed patients, do not include any section about patients with autoimmune inflammatory diseases and patients on immunosuppressive treatment such as biological therapy. Please explain the data available about this special group of patients, including references to the European League against Rheumatism recommendations and reports such as the one by Richi et al “Impact of biological therapies on the immune response after pneumococcal vaccination in patients with autoimmune inflammatory diseases”. Vaccines. 2021, 9, 203. DOI: 10.3390/vaccines903020.
The rest of my comments are minor ones. I have included in the attached text. Please let me know if you have any problem to see them.

Author Response
Manuscript ID: microorganism- 1315860      Date: Aug 10, 2021
Title: Recent topics of pneumococcal vaccination: Indication of pneumococcal vaccine for individuals at a risk of pneumococcal disease in adults
Thank you very much for your great reviewing. We totally agree with your opinions and made appropriate revisions according to your suggestions. The revised portion was red in the text. We strongly believe that this revision made the text better than before. Please, consider my article for publication in your journal.
Sincerely,
Nobuhiro Asai and Hiroshige Mikamo,
Reviewer 3
This in an interesting article by Asai and Mikamo that review the indications of pneumococcal vaccination for individuals at risk of pneumococcal infection. However, it needs some changes and explanations to improve its quality. For this reviewer the main lack is that when authors review the indications in immunosuppressed patients, do not include any section about patients with autoimmune inflammatory diseases and patients on immunosuppressive treatment such as biological therapy. Please explain the data available about this special group of patients, including references to the European League against Rheumatism recommendations and reports such as the one by Richi et al “Impact of biological therapies on the immune response after pneumococcal vaccination in patients with autoimmune inflammatory diseases”. Vaccines. 2021, 9, 203. DOI: 10.3390/vaccines903020.
The rest of my comments are minor ones. I have included in the attached text. Please let me know if you have any problem to see them.
→I agree with your opinion. Thus, I added the section regarding auto immune disease (AIID) as follows. I really apricate your great reviewing and feedback for us.
In page 14
Autoimmune inflammatory diseases
It is well-known that patients with autoimmune inflammatory diseases (AIID) have a risk of pneumococcal disease [34]. The European League against Rheumatism (EULAR) recommendations agree with ACIP guidelines that routine vaccination with PCV13 and PPSV23 for adults aged >19 years with AIID [35]. However, the pneumococcal vaccine coverage rates among AIID patients remain low, showing a percentage of 6 to 62% [36,37]. The main reason for not being vaccinated among the patients is a fear of adverse evets, physicians’ concern that treatment with a biological agent can be associated with an impaired immune response to vaccines [38]. Richi et al. assessed the efficacy on the immune response after pneumococcal vaccination among AIID patients who received a biological agent by using functional antibodies by an opsonophagocytosis killing assay (OPKA) [39]. They showed that AIID patients treated with biological agents had a functional antibody phagocytic response after pneumococcal vaccination using PCV13 and/or PPSV23.
→I revised the text as follows according to your suggestions. Thank for your great help.
In page 2
Abstract
Pneumococcal disease is one of the most common and severe vaccine-preventable diseases (VPDs). Despite the advances in antimicrobial treatment, the pneumococcal disease still remains a global burden and exhibits a high mortality rate among people of all ages worldwide. The immunization program of pneumococcal conjugate vaccine (PCV) in children has decreased pneumococcal disease incidence in several countries. On the contrary, there are several problems regarding the pneumococcal vaccine, such as indications for immunocompetent persons with underlying medical conditions with a risk of pneumococcal disease, the balance of utility and cost, i.e., cost-effectiveness, vaccine coverage rate, serotype replacement, and adverse events. Especially, for individuals aged 19-64 at risk of pneumococcal disease, physicians and vaccine providers should make a rational decision whether the patients should be vaccinated or not, since there is insufficient evidence supporting it. We describe this review regarding topics and problems regarding pneumococcal vaccination from the clinician’s point of view.
In page 3-4
Introduction
Vaccination is one of the most critical public health interventions and has dramatically reduced many kinds of infectious diseases which killed or harmed infants, children, and adults. In many nations, routine immunization programs are provided for infants and children, with fewer countries providing recommendations specifically for adults. Vaccine-preventable diseases (VPDs) remain a significant health burden in infants and children as well as adults. Pneumococcal disease is one of the most common and severe VPDs. Despite the advances in antimicrobial treatments, pneumococcal disease still remains a global burden and a high mortality rate among people of all ages worldwide [1-3]. There are currently, two different types of pneumococcal vaccines such as 13-valent pneumococcal conjugate vaccine (PCV13) and 23-valent pneumococcal polysaccharide vaccine (PPSV23), worldwide. 7-valent pneumococcal conjugate vaccine (PCV7), which is the first vaccine approved for S. pneumoniae, was introduced into the routine pediatric immunization schedule in 2000 and replaced by PCV13 in 2010. In the United States of America, the incidence of PCV13 type disease has been declining to dramatically low levels among people 65 years and older through indirect effects from the pediatric PCV13 program [2]. Although PCV13 is safe and effective for preventing pneumococcal disease among children and people over 65 years, in terms of balancing the evidence and considering acceptability and feasibility, the Advisory Committee on Immunization Practices (ACIP) stated in June 2019 that it no longer routinely recommends PCV13 for all adults aged >65 years and instead, recommends PCV13 based on shared clinical decision-making for adults aged>65years who do not have an immunocompromised condition [2]. Currently, the strategy of pneumococcal vaccination using PCV13 and PPSV23 differs in each country. These have made physicians and vaccine providers confused regarding vaccination. Significantly, the vaccination strategy for 19-64 years with an immunocompetent state is unclear worldwide because of the lack of evidence in this field or the absence of guidelines mentioned in detail. Nevertheless, attending doctors and vaccine providers need to make a rational decision whether the patients should have a vaccination or not. Despite this, there is insufficient evidence about the validity and efficacy of pneumococcal vaccination among immunocompetent patients with a risk of pneumococcal diseases. Besides, the guidelines for each disease also DO NOT mention it in detail. We describe this review from a clinician’s perspective, focusing on these topics that do not have enough evidence justifying vaccination for the patients with a risk of pneumococcal disease.
In page 6
55% (approximately 74 million) of the global infant population not receiving PCV is problematic [8]. Noninvasive pneumococcal disease, including otitis media, sinusitis, and community-acquired pneumonia, are highly prevalent illnesses in children under 5, adults over 65, as well as immunocompromised patients. IPD, such as meningitis, bacteremia, and sepsis, is the most severe presentation and shows high mortality, despite the advancement of antimicrobial treatment. Septic arthritis [9], biliary tract infection [10], infective endocarditis [11], and mycotic aortic aneurysm [12] due to his pathogen has been reported, which sporadically happens. Pneumococcal vaccine is one of the most valuable methods for the prevention of pneumococcal disease.
Zhang et al. reported that by an observational study in the USA, rates of all cause of pneumonia (ACP) and IPD in patients of 19-64 aged with chronic disease and immunocompromised conditions were significantly higher than healthy adults. The rates of ACP and IPD in adults of 19-64 years with chronic disease were 3.6 and 4.6 times higher in healthy adults, respectively, and 5.3 and 10.5 times higher for those with immunocompromised conditions. If the patients have>2 medical conditions, the rates of ACP and IPD were 8.1 and 10.6 times higher for those with chronic disease, and 6.3 and 13.4 times higher for those with immunocompromised conditions in comparison with healthy adults.
I added the following sentence according to your suggestion.
In page 9
Vaccine schedule and interval
For patients who need both pneumococcal vaccines, ACIP recommends that PCV13 is given first, followed by PPSV23 more than 1 year after PCV13 [1]. If PCV has been given, PPSV23 should be given more than 1year after PCV13 and more than 5 years after any PPSV23. If PPSV23 is previously given, PCV13 should be administered at least 1 year after PPSV23. If patients are in special situations such as immunocompromised conditions, cochlear implant, and CSF leak (Table 2), they should have 1 dose PCV13 followed by 1 dose PPSV23 at least 8 weeks later, then another dose of PPSV23 at least 5 years after the previous PPSV23.
I summarized the following sentences according to your suggestion.
In page 11
HIV infected /AIDS
A recent study showed that the incidence rate of pneumococcal disease among patients with human immunodeficiency virus (HIV) infection is 190 per 100,000 patient-years of follow-up, much higher than 38 per 100,000 in the general population [25]. In the study, 224 of the 324 patients (69%) had started Combination Antiretroviral Therapy with a median CD4 T-cell count of 440 cells/mm3 at the diagnosis of pneumococcal disease. Pneumococcal disease among patients with HIV infection often requires hospitalization, and the mortality rates range up to 25% [26]. While ACIP recommends that patients with HIV infection should have pneumococcal vaccination with PCV13, followed 8 weeks later by PPSV23 [1], the clinical efficacy of this combined vaccination schedule is unclear. A systematic review and meta-analysis documented the clinical efficacy of PCV13/PPSV23 among HIV-infected patients [27]. Delaying vaccination until CD4 T-cell count above 200 cells/mm3 is beneficial for vaccination response and should be recommended [27,28].
In page 14
Recommendation for immunocompetent persons with underlying medical conditions
Ref [88,89] were added. I am so surprised that Prof. Nelson E-mailed Ref [88] to me.
In page 21-22
Adverse events by vaccination
Every vaccination could cause adverse events; however, it is not a reason that people hesitate to be vaccinated, except for an anaphylaxis reaction. The most common adverse event of PCV13 is pain in 44.5%, followed by erythema in 27.6% and swelling in 21.0%. For all, 55.6% of the patient getting a shot has some adverse events, which are focal and not severe [87]. Currently, immunization programs using mRNA COVID-19 vaccines have started in the USA. People get vaccinated with mRNA SARS-CoV-2 vaccine. A total of 0.2% of adverse events has been reported. Anaphylaxis reaction happens in only 21/1,000,000 injections (0.000021%). Of those, 17 of 21 had a medical history of drug allergy or anaphylaxis reaction [88]. This result reproduced the previous report [89]. The probability of a severe adverse event associated with vaccination is 0.0007-0.001% [90]. This probability is similar to the odds that a flash of lightning would hit people (0.0001%), or a meteorite could hit and kill people (0.0006%) [91,92]. It would be strange to hesitate to get a vaccine for such a low probability of adverse events. Clinicians should be aware of this fact and provide patients requiring a vaccine with correct information.
I removed the following portions in the sentence.
In page 23-24
Surprisingly, the VCR of the pneumococcal vaccine is much lower than we expect [36]. Unfortunately, there are no data regarding VCR among patients aged 19-64 years, so we could not find why the pneumococcal vaccine has not prevailed widely. We need a further study to analyze this in the population. The potential solutions to improve the low VCR in the population include not only an education to the patients about the importance of vaccination but also reminders for health care professionals, as well as follow-up protocols with patients to make sure they have completed the vaccines. (, and serologic tests are routinely checked and repeated when appropriate.) was removed.
